

# Association between *IL-1B (-511)/IL-1RN* (VNTR) polymorphisms and type 2 diabetes: a systematic review and meta-analysis

Juan Jiao[1,2,*], Zhaoping Wang[1,*], Yanfei Guo[3], Jie Liu[2], Xiuqing Huang[1], Xiaolin Ni[1], Danni Gao[1,4], Liang Sun[1], Xiaoquan Zhu[1], Qi Zhou[1], Ze Yang[1] and Huiping Yuan[1]

[1] The Key Laboratory of Geriatrics, Beijing Institute of Geriatrics, Institute of Geriatric Medicine, Chinese Academy of Medical Science, Beijing Hospital/National Center of Gerontology of National Health Commission, P.R. China
[2] Department of Clinical Laboratory, the Seventh Medical Center, Chinese PLA General Hospital, Beijing, China
[3] Department of Respiratory and Critical Care Medicine, Beijing Hospital, National Center of Gerontology; Institute of Geriatric Medicine, Chinese Academy of Medical Sciences, P.R. China
[4] Peking University Fifth School of Clinical Medicine (Beijing Hospital), Beijing, China
[*] These authors contributed equally to this work.

Corresponding author
Huiping Yuan,
yuanhuiping@126.com

## ABSTRACT

Interleukin-1 (IL-1) plays an essential role in the immune pro-inflammatory process, which is regarded as one of many factors in the development of type 2 diabetes mellitus (T2DM). Several case-control studies have illustrated the association of the *IL-1B (-511)* (rs16944, Chr 2:112,837,290, C/T Intragenic, Transition Substitution) and *IL-1RN* (VNTR) (gene for IL-1 receptor antagonist, IL-1RA, 86 bp tandem repeats in intron 2) polymorphisms with T2DM risk. However, the results were inconsistent and inconclusive. We performed a meta-analysis (registry number: CRD42021268494) to assess the association of the *IL-1B (-511)* and *IL-1RN* (VNTR) polymorphisms with T2DM risk. Random-effects models were applied to calculate the pooled ORs (odds ratios) and 95% CIs (confidence intervals) to test the strength of the association in the overall group and subgroups stratified by ethnicity, respectively. Between-study heterogeneity and publication bias were evaluated by the $Q$-test, $I^2$ statistic, Harbord test, and Peters test accordingly. Sensitivity analyses were also performed. A total of 12 publications evaluating the association of *IL-1B (-511)* and *IL-1RN* (VNTR) polymorphisms with the risk of T2DM development were included. The meta-analysis showed that *IL-1RN* (VNTR) was related to the increasing development of T2DM risk in the recessive model (OR = 1.62, 95% CI [1.09–2.42], $P_{het} = 0.377$, $P_z = 0.018$) and in the homozygous model (OR = 2.02, 95% CI [1.07–3.83], $P_{het} = 0.085$, $P_z = 0.031$), and the *IL-1RN 2\** allele was found a significant association with evaluated T2DM risk in all ethnicities (OR = 2.08, 95% CI [1.43–3.02], $P_{het} < 0.001$, $P_z < 0.001$) and in EA (OR = 2.01, 95% CI [1.53–2.66], $P_{het} = 0.541$, $P_z < 0.001$). Moreover, stratification by ethnicity revealed that *IL-1B (-511)* was associated with a decreased risk of T2DM in the dominant model (OR=0.76, 95% CI [0.59–0.97], $P_{het} = 0.218$, $P_z = 0.027$) and codominant model (OR = 0.73, 95% CI [0.54–0.99], $P_{het} = 0.141$, $P_z = 0.040$) in the East Asian (EA) subgroup. Our results suggest that the *IL-1RN 2\** allele and *2\*2\** homozygous polymorphism are strongly associated with increasing T2DM risk and that

the *IL-1B (-511) T* allele polymorphism is associated with decreasing T2DM risk in the EA subgroup.

## INTRODUCTION

The latest data from the 9th edition of the IDF Diabetes Atlas show that approximately 463 million adults (20–79 years) have diabetes, and by 2045, the number will rise to 700 million. Type 2 diabetes mellitus (T2DM) is the most prevalent type of diabetes (>90% of diabetes). The proportion of T2DM patients is increasing in most countries, especially in low- and middle-income countries such as China and India (https://www.idf.org/aboutdiabetes/what-is-diabetes/facts-figures.html). Growing evidence suggests that the risk of T2DM is associated with various factors, such as genetics, ethnicity, environment, and lifestyle. In particular, genetic susceptibility seems to play an essential role in the pathogenesis of T2DM (*Hivert, Vassy & Meigs, 2014*).

T2DM occurs when β-cell function progressively deteriorates and fails to compensate for insulin resistance, partly due to the demise of pancreatic β-cells through apoptosis (*Donath & Halban, 2004*). The chronic and low-grade inflammation in metabolic organs including the liver, brain, pancreas and adipose tissue is known as metaflammation (metabolism-induced inflammation), which can be widely observed in T2DM (*Hotamisligil, 2006*; *Hotamisligil, 2017*). Increasing evidence has shown that metaflammation plays a vital role in the development of T2DM and its cardiovascular complications (*Hotamisligil, 2017*) and pro-inflammatory cytokines, especially interleukin-1β (IL-1β), play a key role in this process (*Dinarello, Donath & Mandrup-Poulsen, 2010*; *Fève & Bastard, 2009*; *Sathyapalan & Atkin, 2011*). Moreover, the increased level of IL-1β in human pancreatic cells due to the elevated glucose concentrations and decreased level of IL-1 receptor antagonist (IL-1Ra) in islets of T2DM patients result in impaired insulin secretion, decreased cell proliferation, and apoptosis of β-cells (*Böni-Schnetzler & Donath, 2013*). A previous observational study demonstrated that a combined elevation of IL-1β and IL-6 was associated with a roughly threefold increased risk of T2DM, and IL-1β might induce insulin resistance *via* activating the IκB kinaseβ (*Spranger et al., 2003*). Furthermore, CANTOS and TRACK trials have evaluated the role of IL-1 inhibition in the treatment of several inflammatory disorders, such as coronary artery disease, T2DM and rheumatoid arthritis (*Qamar & Rader, 2012*; *Ruscitti et al., 2019*). According to the result from CANTOS trials, canakinumab (a fully humanized monoclonal antibody) could selectively block IL-1β, and its efficacy in suppressing both levels of inflammatory markers and clinical symptoms in patients with autoinflammatory disease such as T2DM and impaired glucose tolerance has been observed in this trial (*Aday & Ridker, 2018*; *Rissanen et al., 2012*). Therefore, the genetic polymorphisms that regulate the expression levels of IL-1β and IL-1Ra might have an essential impact on the interindividual differences in T2DM.

Consisting of three linked genes and mapping to chromosome 2q13-24, IL-1 cytokine genes encode IL-1α, IL-1β and IL-1Ra. Both IL-1α and IL-1β are pro-inflammatory cytokines, while IL-1Ra can inhibit inflammation by competing for receptor binding (*Dinarello, 2000*). In the promoter region, *IL-1B (-511)* (rs16944, Chr 2:112,837,290, C/T Intragenic, Transition Substitution) has a bi-allelic polymorphism at position -511, representing the C/T transition (*Kristiansen et al., 2000*). *IL-1RN* (VNTR) (gene for IL-1 receptor antagonist, IL-1RA, 86 bp tandem repeats in intron 2) has five different alleles: *1\** (four repeats), *2\** (two repeats), *3\** (five repeats), *4\** (three repeats) and *5\** (six repeats) (*Tarlow et al., 1993*). *IL-1RN 1\** (four repeats) and *IL-1RN 2\** (two repeats) are the most common, whereas the others occur much less frequently (<5%) (*Santtila, Savinainen & Hurme, 1998*). Many genetic association studies have been performed to estimate the relationship of the *IL-1B (-511)* and *IL-1RN* (VNTR) polymorphisms with T2DM risk (*Achyut et al., 2007*; *Muktabhant et al., 2013*). Several results suggest that there is no relationship between them (*Borilova Linhartova et al., 2019*; *Muktabhant et al., 2013*), whereas *Achyut et al., (2007)* showed that both the *IL-1B (-511)* and *IL-1RN* (VNTR) polymorphisms were associated with susceptibility to T2DM as well as complications, and he indicated that *T2* (*IL-1β −511T/IL-1RN\*2*) haplotype was associated with a roughly twofold increased risk of T2DM. Considering the small sample sizes ($n < 600$) and varying population characteristics (ethnic differences) in different studies, the results are conflicting. To illustrate the potential association of the *IL-1B (-511)* and *IL-1RN* (VNTR) polymorphisms with T2DM risk, we conducted a meta-analysis including 12 reported publications.

## MATERIALS AND METHODS

This systematic review was conducted following the Preferred Reporting Items for Systematic Reviews and Meta-Analysis (PRISMA) statement guidelines (*Moher et al., 2009*). A protocol was registered before commencing this review on PROSPERO (CRD42021268494).

### The search strategy

The online databases we searched included PubMed, Web of Science, CNKI (China National Knowledge Infrastructure) and Wanfang. We identified relevant articles reporting the association of *IL-1RN* (VNTR) and *IL-1B (-511)* polymorphisms with the risk of T2DM in the medical literature through the end of September 12, 2020. This study used a combination of the following search terms: "Type 2 Diabetes Mellitus" or "T2DM" or "interleukin-1" or "IL-1" or "polymorphism" or "genetics" or "association". In addition, we also conducted a hand search to identify relevant data in references from retrieved articles. Juan Jiao and Zhaoping Wang performed the Search Strategy. In case of disagreement, it will be settled by a third assessor's evaluation and discussed until a consistent result was reached.

### The inclusion and exclusion criteria

All studies included in this analysis met the following criteria: (1) the study revealed the relationship of *IL-1B (-511)* and *IL-1RN* (VNTR) with T2DM risk; (2) the design

was a "case-control study"; (3) the research subjects were humans with T2DM; (4) the publication language was English or Chinese; and (5) the study provided sufficient data to estimate ORs (odds ratios) and 95% CIs (confidence intervals). The major exclusion criteria were as follows: (1) duplicate data; (2) randomly chosen controls; (3) non-research articles; and (4) insufficient reporting of data.

## Data extraction

All data were systematically reviewed and extracted by two investigators (ZPW and DNG) according to a standardized form, and then all of the following information was collected in an electronic database: the first author's name, year of publication, country of origin, ethnicity, total number of cases/controls, genotyping method, diagnostic criteria of T2DM, genotypic frequencies and language of the report. Disagreement was settled by a third assessor's evaluation and discussed until a consistent conclusion was agreed.

## Assessment of quality score

We assessed the quality score of identified publications based on the Newcastle Ottawa Scale (NOS) (http://www.ohri.ca/programs/clinical_epidemiology/oxford.asp). And the NOS involved "selection", "comparability", and "estimation of outcomes or exposures". The scores ranged from 0 to 9, and the score of "high quality study" was $\geq 6$ (detailed scores see Table S1) (*Lo, Mertz & Loeb, 2014*). And the STREGA (Strengthening the REporting of Genetic Association) system was also performed to assessed the methodological quality of all included studies, which includes 22 items with scoring from 0 to 22 (detailed scores see Table S2) (*Little et al., 2009*). And the score of high quality was 18–22; the score of moderate-high quality was 13–17; the score of low quality was 0–12 (*Duan et al., 2018*).

## Data analysis

We conducted this analysis and entered data with Stata software, version 15.0 (Stata Corp., College Station, TX, USA). We used original genotypic distribution data without any adjustment to measure the strength of the association between *IL-1B (-511)/IL-1RN* and T2DM risk by ORs (odds ratios) with 95% CIs (confidence intervals) under dominant, recessive, additive (overdominant+codominant) and homozygous models. The distributions of genotype frequencies of controls were all consistent with Hardy-Weinberg Equilibrium (*HWE*) ($P > 0.05$, Table 1). Between-study heterogeneity among the studies was tested with Cochran's $Q$ and the $I^2$ test statistics. $I^2$ was calculated based on the formula $I^2 = 100\% \times (Q - df)/Q$. Heterogeneity was regarded as significant when $P_{het}$ < 0.1 or $I^2 > 50\%$. For those with significant heterogeneity, we performed stratified analyses to examine the statistical significance of the difference in ORs according to ethnicity (East Asian, South Asian, North African and Caucasian). We used the random-effects model (DerSimonian and Laird method) which was combined by applying inverse variance-weighted meta-analysis to calculate the pooled OR and 95% CI, with $P_z$ < 0.05 considered statistically significant (*DerSimonian & Laird, 1986*). When the heterogeneity was significant, sensitivity analyses were conducted to evaluate the influence of each single study by omitting one study at a time and checking the pool effect size for the remainder of the studies. Finally, the Harbord test and Peters test were used when fewer than ten

Jiao et al. (2021), *PeerJ*, DOI 10.7717/peerj.12384

**Table 1  Characteristics of Identified Studies on *IL-1B (-511)/IL-1RN(VNTR)* polymorphism and the risks of T2DM.**

| Author | Year | Country/Ethnicity | Diagnostic criteria | Genotyping method | Number | | Language | IL-1 gene | QA | HWE |
|---|---|---|---|---|---|---|---|---|---|---|
| | | | | | Cases | Controls | | | | |
| Zhang Jian | 2004 | China/EA | NA | PCR-RFLP | 106 | 198 | Chinese | *IL-1RN* | 7 | 1.00 |
| Zhang Ping-an | 2004 | China/EA | WHO, 1999 | PCR-RFLP | 106 | 247 | Chinese | *IL-1RN* | 8 | 0.70 |
| Zhou Jian-zhong | 2010 | China/EA | WHO, 1999 | PCR-RFLP | 72 | 97 | Chinese | *IL-1RN* | 7 | 0.23 |
| Petra Borilova Linhartova | 2019 | Czech Republic/Caucasian | NA | PCR-RFLP | 380 | 212 | English | *IL-1RN* | 8 | 1.00 |
| Alexandra I. F. Blakemore | 1995 | North of England/Caucasian | WHO, NA | PCR | 117 | 248 | English | *IL-1RN* | 7 | 0.61 |
| B.R Achyut | 2006 | North Indian/SA | WHO, 1999 | PCR-RFLP | 200 | 223 | English | *IL-1RN* *IL-1B (-511)* | 8 | 0.100.16 |
| Liu Chang | 2014 | China/EA | WHO, 1999 | PCR-RFLP | 583 | 366 | Chinese | *IL-1B (-511)* | 7 | 1.00 |
| Cao Yong | 2013 | China/EA | WHO, 1999 | PCR-RFLP | 268 | 263 | Chinese | *IL-1B (-511)* | 8 | 0.06 |
| Lin Neng-bo | 2016 | China/EA | WHO, 1999 | PCR-RFLP | 286 | 327 | Chinese | *IL-1B (-511)* | 8 | 0.08 |
| Natalie E. Doody | 2017 | North Indian/SA | WHO, 1999 | PCR-RFLP | 202 | 203 | English | *IL-1B (-511)* | 8 | 0.87 |
| Benja Muktabhant | 2013 | Thailand/SEA | NA | PCR-RFLP | 90 | 30 | English | *IL-1B (-511)* | 7 | 0.16 |
| Safaa I. Tayel | 2018 | Egypt/NA$_1$ | ADA, NA | TaqMan | 50 | 30 | English | *IL-1B (-511)* | 8 | 1.00 |

**Notes.**

T2DM, Type 2 diabetes mellitus; EA, East Asian; SA, South Asian; SEA, South East Asian; NA, Not Available (the diagnosis of T2DM was originally based on the presence of clinical symptoms and biochemical); NA1, North African; ADA, American Diabetes Association (fasting blood glucose ≥126 mg/dL or 2 h blood glucose after overload with 75 g of glucose ≥ 200 mg/dL in oral glucose tolerance test (OGTT) or glycated hemoglobin (HbA1c) ≥6.5 in patients with classic symptoms of hyperglycemia); WHO, World Health Organization (fasting glucose levels [7.0 mmol/L or 126 mg/dL] after a minimum 12-h fast or 2-h post glucose level (oral glucose tolerance test or 2-h OGTT) [11.1 mmol/L or 200 mg/dL] on more than one occasion); QA, Quality Assessment; HWE, Hardy-Weinberg Equilibrium.

articles were included, as they are more sensitive for assessing publication bias, with $P < 0.1$ assuming that the bias was statistically significant.

## RESULTS

### Included studies

Figure 1 shows the detailed process of selecting and assessing eligible studies. We identified 450 publications *via* the initial keyword search. After screening, 12 publications met our inclusion criteria (*Achyut et al., 2007*; *Blakemore et al., 1996*; *Borilova Linhartova et al., 2019*; *Cao et al., 2013*; *Doody et al., 2017*; *Lin et al., 2016*; *Liu et al., 2014*; *Muktabhant et al., 2013*; *Tayel et al., 2018*; *Zhang, Xiao & Li, 2004a*; *Zhang et al., 2004b*; *Zhou et al., 2010*), and 438 studies were excluded for containing duplicate data, missing critical data, and being review papers or studies that were not related to T2DM. Table 1 illustrates the characteristics of all included publications in this meta-analysis. In terms of polymorphisms and diseases, six identified case-control studies, comprising 981 cases and 1,225 controls, regarded *IL-1RN* (VNTR) to evaluate its association with T2DM risk (*Achyut et al., 2007*; *Blakemore et al., 1996*; *Borilova Linhartova et al., 2019*; *Zhang, Xiao & Li, 2004a*; *Zhang et al., 2004b*; *Zhou et al., 2010*), and seven identified case-control studies, including 1,679 cases and 1,442 controls, tested the association between *IL-IB (-511)* and T2DM risk (*Achyut et al., 2007*; *Cao et al., 2013*; *Doody et al., 2017*; *Lin et al., 2016*; *Liu et al., 2014*; *Muktabhant et al., 2013*; *Tayel et al., 2018*). Stratified by ethnicity, six publications were performed in East Asian (EA) populations (*Cao et al., 2013*; *Lin et al., 2016*; *Liu et al., 2014*; *Zhang, Xiao & Li, 2004a*; *Zhang et al., 2004b*; *Zhou et al., 2010*), two studies were undertaken in South Asian (SA) populations (*Achyut et al., 2007*; *Doody et al., 2017*), two studies were conducted in Caucasian populations (*Blakemore et al., 1996*; *Borilova Linhartova et al., 2019*), and the remaining studies were completed in Southeast Asian (SA) (*Muktabhant et al., 2013*) and North African (NA$_1$) populations (*Tayel et al., 2018*).

### Individual polymorphism meta-analysis

The genotypic distributions of the *IL-1B (-511)* and *IL-1RN* (VNTR) polymorphisms are shown in Table 2. Table 3 shows the meta-analysis results for these two polymorphisms.

For the *IL-1B (-511)* polymorphism, seven studies with 1,679 cases and 1,442 controls were included in the meta-analysis. The results of the pooled analysis suggest that the *IL-1B (-511)* polymorphism is not significantly associated with T2DM risk in all study subjects under the dominant model (OR = 0.84, 95% CI [0.57–1.25], $P_{het} = 0.001$, $P_z = 0.395$) (Table 3 and Fig. 2A), recessive model (OR = 0.89, 95% CI [0.61–1.31], $P_{het} < 0.001$, $P_z = 0.561$) (Table 3), and homozygous model (OR = 0.82, 95% CI [0.48–1.39], $P_{het} < 0.001$, $P_z = 0.451$) (Table 3). In the stratification analyses by ethnicity, we found that there was a significant association between the *IL-1B (-511)* polymorphism and decreasing T2DM risk in the EA population under the dominant model (OR = 0.76, 95% CI [0.59–0.97], $P_{het} = 0.218$, $P_z = 0.027$) (Table 3 and Fig. 2A) and codominant model (OR = 0.73, 95% CI [0.54–0.99], $P_{het} = 0.141$, $P_z = 0.040$) (Table 3), but no association was observed under the recessive model (OR = 0.97, 95% CI [0.79–1.20], $P_{het} = 0.589$, $P_z = 0.789$) or homozygous model (OR = 0.80, 95% CI [0.62–1.03], $P_{het} = 0.571$, $P_z = 0.081$). In

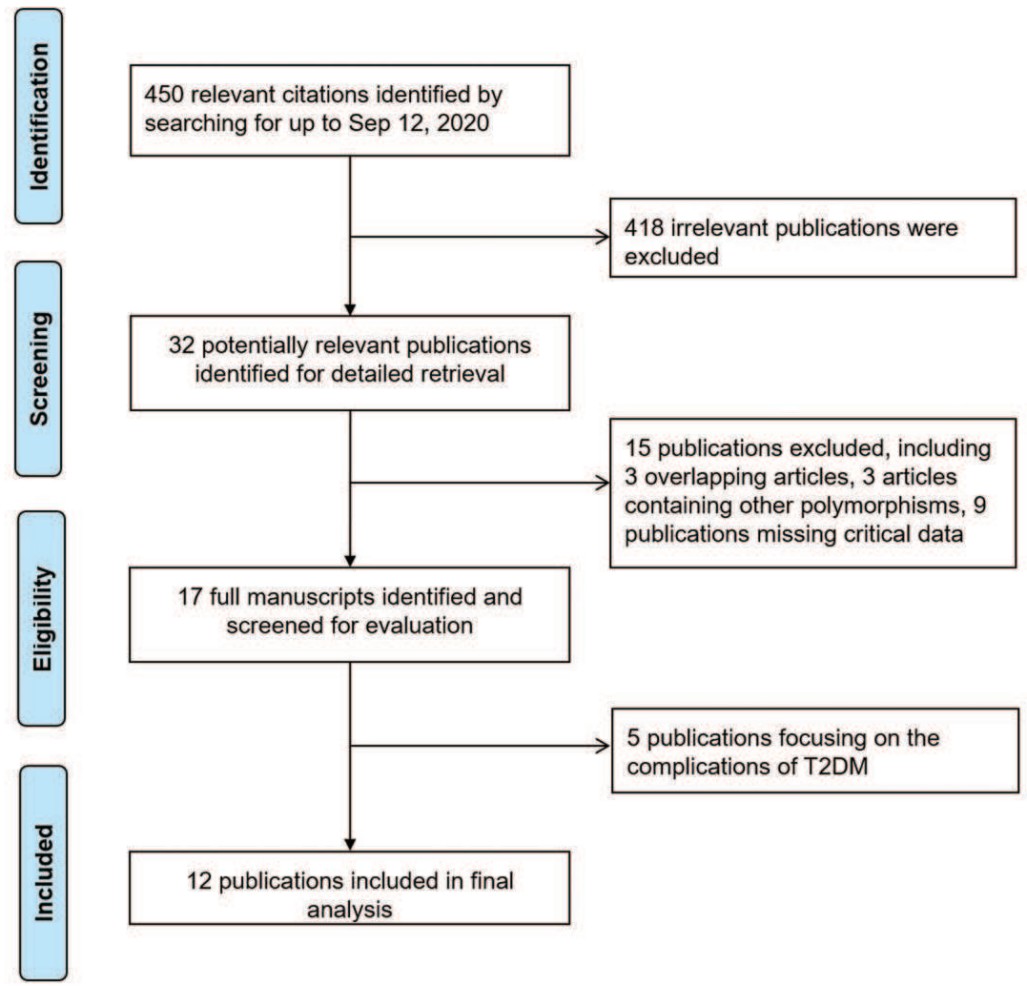

**Figure 1** Flowchart of study identification process.

other ethnic groups, no association between the *IL-1B (-511)* polymorphism and T2DM risk was found. Between-study heterogeneity was found in the dominant, recessive and homozygous models ($P_{het} = 0.001$, $P_{het} < 0.001$, $P_{het} < 0.001$) (Table 3).

For the *IL-1RN* (VNTR) polymorphism, six publications were involved in the meta-analysis (980 cases and 1,225 controls). Overall, a significantly increased T2DM risk was found to be associated with *2\** allele in all ethnicities (OR = 2.08, 95% CI [1.43–3.02], $P_{het} < 0.001$, $P_z < 0.001$) and in EA (OR = 2.01, 95% CI [1.53–2.66], $P_{het} = 0.541$, $P_z < 0.001$) (Table 3). Pooling data also revealed that this polymorphism is strongly associated with an increasing T2DM risk in the recessive model (OR = 1.62, 95% CI [1.09–2.42], $P_{het} = 0.377$, $P_z = 0.018$) (Table 3 and Fig. 3A) and homozygous model (OR = 2.02, 95% CI [1.07–3.83], $P_{het} = 0.085$, $P_z = 0.031$) (Table 3 and Fig. 3B), but there was no association in the dominant model (OR = 1.35, 95% CI [0.79–2.31], $P_{het} < 0.001$, $P_z = 0.275$) (Table 3). Between-study heterogeneity was found in the dominant model ($P_{het} < 0.001$) (Table 3).

**Table 2  The IL-1 polymorphism distribution in cases and controls.** Table 2 shows the genotypic distributions of the *IL-1B (-511)* and *IL-1RN (VNTR)* polymorphisms.

| Gene | Case | | | | | Control | | | | |
|---|---|---|---|---|---|---|---|---|---|---|
| *IL-1B (-511)* | *C* | *T* | *CC* | *TC* | *TT* | *C* | *T* | *CC* | *TC* | *TT* |
| Cao Yong | 302 | 270 | 78 | 129 | 61 | 332 | 322 | 52 | 147 | 64 |
| Liu Chang | 634 | 532 | 171 | 292 | 120 | 385 | 347 | 101 | 183 | 82 |
| B.R Achyut | 108 | 292 | 9 | 90 | 101 | 175 | 271 | 29 | 117 | 77 |
| Lin Neng-bo | 302 | 270 | 85 | 132 | 69 | 332 | 322 | 76 | 180 | 71 |
| Safaa I. Tayel | 55 | 45 | 16 | 23 | 11 | 21 | 39 | 4 | 13 | 13 |
| Natalie E. Doody | 186 | 218 | 40 | 106 | 56 | 133 | 273 | 21 | 91 | 91 |
| Benja Muktabhant | 90 | 90 | 23 | 44 | 23 | 33 | 27 | 11 | 11 | 8 |
| *IL-1RN* | *1** | *2** | *1*/1** | *1*/2** | *2*/2** | *1** | *2** | *1*/1** | *1*/2** | *2*/2** |
| B.R Achyut | 227 | 234 | 60 | 107 | 20 | 340 | 92 | 138 | 64 | 14 |
| Zhang Jian | 192 | 34 | 88 | 16 | 2 | 360 | 36 | 163 | 34 | 1 |
| Zhang Ping-an | 192 | 34 | 88 | 16 | 2 | 450 | 44 | 204 | 42 | 1 |
| Zhou Jian-zhong | 107 | 71 | 36 | 35 | 1 | 153 | 41 | 58 | 37 | 2 |
| Alexandra I. F. Blakemore | 154 | 106 | 55 | 44 | 18 | 372 | 124 | 141 | 90 | 17 |
| Petra Borilova Linhartova | 569 | 289 | 220 | 129 | 31 | 303 | 121 | 108 | 87 | 17 |

## Sensitivity and publication bias analysis

We next performed sensitivity analysis on the association between IL-1β (-511) and T2DM. The results showed that Achyut's study had an influence on the pooled OR in the IL-1β (-511)-dominant model (OR = 0.71, 95% CI [0.53–0.96]) and homozygous model (OR = 0.65, 95% CI [0.44–0.97]) (Figs. 2B, 2D) but did not affect the pooled OR in the recessive model (Fig. 2C). Achyut's study was performed in a SA population; thus, there was no influence on the pooled OR in the EA population. Due to the lack of heterogeneity in IL-1RN recessive and homozygous models, sensitivity analysis was not performed.

The results of the Harbord test and Peters test showed no publication bias for *IL-1B (-511)* in the dominant model ($P_{har} = 0.759$ and $P_{pet} = 0.881$, respectively) (Fig. 4A). Moreover, no evidence of publication bias was found for *IL-1RN* (VNTR) in the recessive ($P_{har} = 0.498$ and $P_{pet} = 0.495$, respectively) or homozygous models ($P_{har} = 0.634$ and $P_{pet} = 0.481$, respectively) (Figs. 4B, 4C).

## DISCUSSION

The *IL-1B (-511)* and *IL-1RN* (VNTR) polymorphisms have been reported to be related to the pathogenesis of T2DM (*Ehses et al., 2009*; *Masters et al., 2010*), and many case-control studies have illustrated the association of *IL-1B (-511)* and *IL-1RN* (VNTR) polymorphisms with T2DM risk. However, each individual study may not have been powerful, causing their results to be controversial. Given the above, we conducted the present meta-analysis to derive a more precise evaluation of the association of *IL-1B (-511)* and *IL-1RN* (VNTR) polymorphisms with T2DM risk. We found that (i) the *T* allele of *IL-1B (-511)* was related with a decreased T2DM risk in the EA subgroup; (ii) significantly elevated risk of developing T2DM was observed to be associated with the *2** allele in all ethnicities and

**Table 3  Meta-analysis of the relationship of the IL-1 polymorphisms with the risks of T2DM.**

| Gene | No. studies | Allele comparison | | | Genetic model comparison | | | |
|---|---|---|---|---|---|---|---|---|
| | | OR (95% CI) | $P_{het}$ | $P_z$ | OR (95% CI) | | $P_{het}$ | $P_z$ |
| *IL-1B (-511)* | | | | | | | | |
| Total | 7 | 0.89 (0.68–1.17) | <0.001 | 0.406 | Dominant | 0.84 (0.57–1.25) | 0.001 | 0.395 |
| | | | | | Recessive | 0.89 (0.61–1.31) | <0.001 | 0.561 |
| | | | | | Overdominant | 0.92 (0.74–1.14) | 0.065 | 0.422 |
| | | | | | Codominant | 0.85 (0.6–1.21) | 0.011 | 0.375 |
| | | | | | Codominant[#] | 0.96 (0.69–1.34) | 0.003 | 0.803 |
| | | | | | Homozygote | 0.82 (0.48–1.39) | <0.001 | 0.451 |
| EA | 3 | 0.89 (0.79–1.01) | 0.606 | 0.072 | **Dominant** | **0.76 (0.59–0.97)** | **0.218** | **0.027** |
| | | | | | Recessive | 0.97 (0.79–1.20) | 0.589 | 0.789 |
| | | | | | Overdominant | 0.82 (0.64–1.04) | 0.162 | 0.094 |
| | | | | | **Codominant** | **0.73 (0.54–0.99)** | **0.141** | **0.040** |
| | | | | | Codominant[#] | 1.07 (0.86–1.34) | 0.385 | 0.531 |
| | | | | | Homozygote | 0.80 (0.62–1.03) | 0.571 | 0.081 |
| SA | 2 | 1.00 (0.33–2.98) | <0.001 | 0.997 | Dominant | 1.20 (0.18–7.84) | <0.001 | 0.853 |
| SEA | 1 | 1.22 (0.68–2.20) | – | 0.503 | Dominant | 1.69 (0.70–4.07) | – | 0.245 |
| NA | 1 | **0.44 (0.23–0.85)** | – | **0.015** | Dominant | 0.33 (0.10–1.10) | – | 0.070 |
| *IL-1RN* | | | | | | | | |
| Total | 6 | **2.08 (1.43–3.02)** | **<0.001** | **<0.001** | Dominant | 1.35 (0.79–2.31) | <0.001 | 0.275 |
| | | | | | **Recessive** | **1.62 (1.09–2.42)** | **0.377** | **0.018** |
| | | | | | Overdominant | 1.20 (0.72–1.99) | <0.001 | 0.485 |
| | | | | | Codominant | 1.28 (0.73–2.24) | <0.001 | 0.398 |
| | | | | | Codominant[#] | 1.38 (0.89–2.13) | 0.351 | 0.155 |
| | | | | | **Homozygote** | **2.02 (1.07–3.83)** | **0.085** | **0.031** |
| EA | 3 | **2.02 (1.53–2.66)** | **0.541** | **<0.001** | Dominant | 1.11 (0.78–1.59) | 0.525 | 0.558 |
| | | | | | Recessive | 2.30 (0.57–9.25) | 0.470 | 0.243 |
| | | | | | Overdominant | 1.05 (0.72–1.54) | 0.337 | 0.790 |
| | | | | | Codominant | 1.06 (0.74–1.53) | 0.371 | 0.749 |
| | | | | | Codominant[#] | 2.25 (0.53–9.55) | 0.356 | 0.270 |
| | | | | | Homozygote | 2.42 (0.60–9.78) | 0.554 | 0.216 |
| SA | 1 | **3.81 (2.84–5.11)** | – | **<0.001** | Dominant | 3.75 (2.48–5.67) | – | <0.001 |
| Caucasian | 2 | 1.60 (0.998–2.58) | 0.02 | 0.051 | Dominant | 1.04 (0.54–2.02) | 0.017 | 0.901 |

**Notes.**

CI, confidence interval; IL-1, interleukin-1; OR, odds ratio; $P_{het}$, P-value for heterogeneity; $P_z$, P-value for overall effect; SA, South Asian; SEA, South East Asian; EA, East Asian; NA, North Africa.

For *IL1B (-511)* polymorphism: dominant (*TT + TC vs CC*), recessive (*TT vs TC + CC*), overdominant (*CT vs CC+TT*), codominant (*CT vs CC*), codominant [#] (*CT vs TT*) and homozygote (*TT vs CC*).

For *IL1RN (VNTR)* polymorphism: dominant (*2\*2\* + 2\*1\* vs 1\*1\**), recessive (*2\*2\* vs 2\*1\* + 1\*1\**), overdominant (*2\*1\* vs 1\*1\*+2\*2\**), codominant (*2\*1\* vs 1\*1\**), codominant [#] (*2\*1\*vs 2\*2\**) and homozygote (*2\*2\* vs 1\*1\**).

EA subgroup, and its carriers of the *IL-1RN* (VNTR) polymorphism had an increased risk of developing T2DM among all ethnicities; (iii) compared to the *1\*/1\** homozygotes, the *2\*/2\** homozygotes of the *IL-1RN* (VNTR) polymorphism had an increased risk of T2DM development; and (iv) no significant association was observed between the *IL-1RN* (VNTR) polymorphism and T2DM risk in the EA subgroup. The results of sensitivity
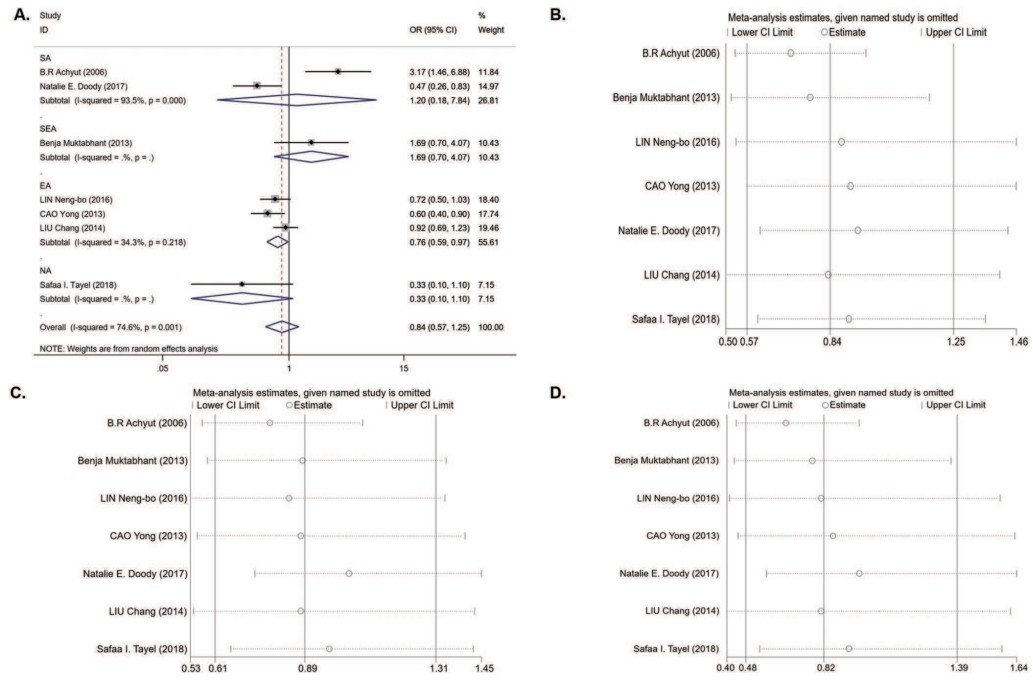

**Figure 2** An overall forest plot of *IL-1B (-511)* associated with T2DM risk in the dominant model (*TT + TC vs CC*) and sensitivity analyses. (A) Forest plot of *IL-1B (-511)* associated with T2DM in the dominant model; (SA: South Asian, SEA: South East Asian, EA: East Asian, NA: North Africa). (B) Sensitivity analysis for *TT + TC vs CC* associated with T2DM in the overall meta-analysis. (C) Sensitivity analysis for *TT vs TC + CC* associated with T2DM in the overall meta-analysis. (D) Sensitivity analysis for *TT vs CC* associated with T2DM in the overall meta-analysis.

analysis showed that Achyut's study affected the pooled ORs in the IL-1β (-511) dominant and homozygous models. However, Achyut's study did not influence the pooled OR in the EA subgroup after stratification by ethnicity. Therefore, the result of the association between *IL-1B (-511)* and the decreased risk of T2DM in the dominant model in the EA subgroup was convincing.

First, IL-1β was considered a mediator of fever, and rapidly, it was found that IL-1β induced innate immunity to defend against pathogens. However, chronic overexpression of IL-1β has been related to multiple immune diseases, including T2DM (*Mandrup-Poulsen, 1996*). Eventually, it was suggested that IL-1β can induce the inflammatory microenvironment of islets, leading to impaired insulin secretion, decreased cell proliferation and apoptosis of β-cells, ultimately contributing to the development of T2DM (*Rhodes, 2005*). Donath, M Y et al. extensively described the development of islet inflammation in GK rats, suggesting that the expression of pro-inflammatory cytokines IL-1β and others (IL-6, TNFα) was increased in islets, while elevated expression levels of many chemokines (CXCL1/KC, MCP-1, MIP-1α) and infiltration of immune cells in islets were observed (*Ehses et al., 2009*). In all cases, macrophage infiltration is increased in the islet inflammatory process (*Ehses et al., 2007*). Moreover, two independent studies have now indicated that the numbers of islet-related CD68[+] cells are increased in T2DM patients

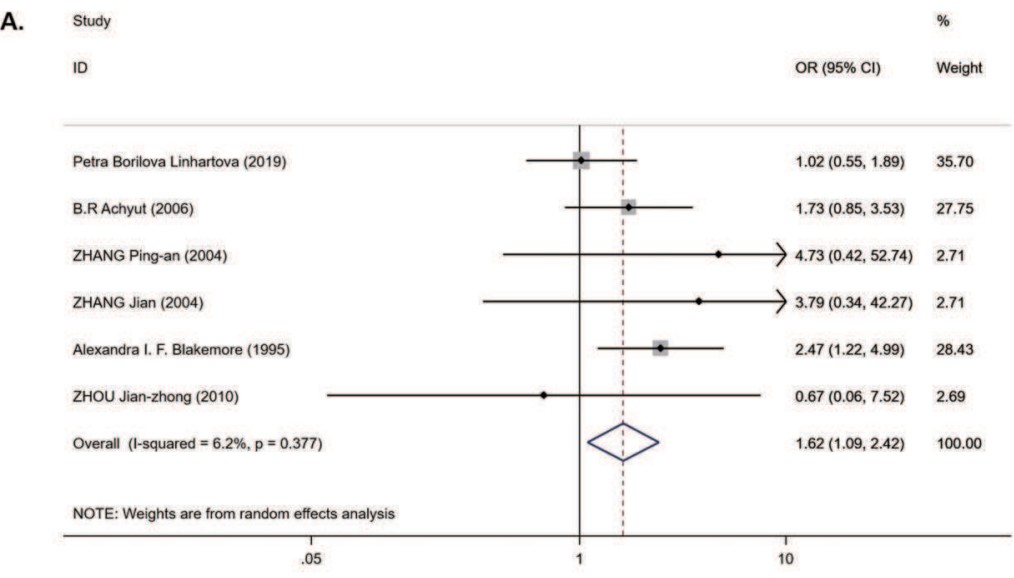

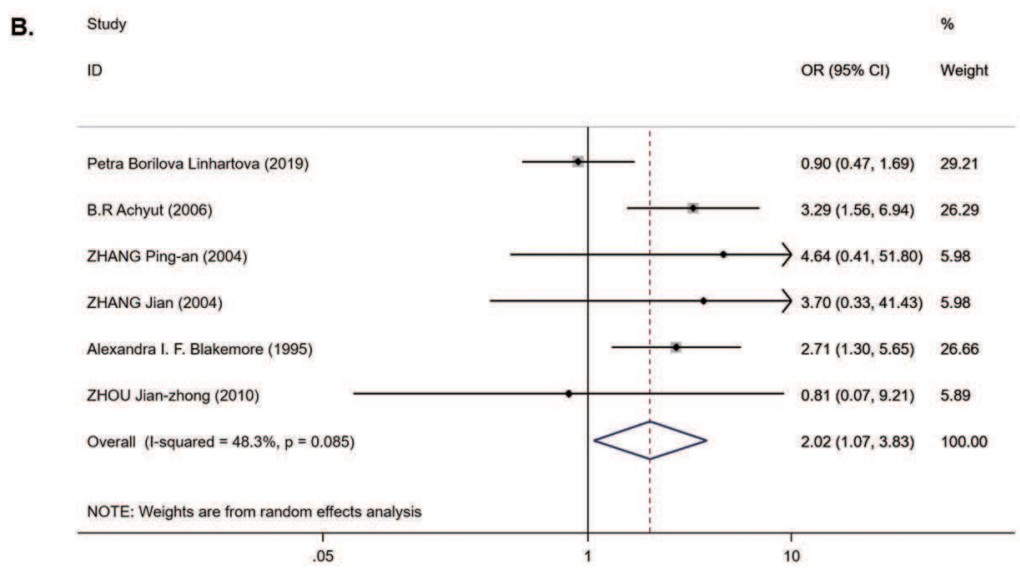

**Figure 3** **Forest plot of meta-analysis on *IL-1RN* (VNTR) and the risk of T2DM in the recessive model** (*2\*2\* vs 2\*1\* + 1\*1\**) **and homozygous model (*2\*2\* vs 1\*1\**).** (A) Forest plot of *IL-1RN* (VNTR) associated with T2DM in the recessive model. (B) Forest plot of *IL-1RN* (VNTR) associated with T2DM in the homozygous model.

(*Ehses et al., 2007*; *Richardson et al., 2009*). In addition, data obtained from laser-captured β-cells from T2DM patients have shown that evaluated expression levels of IL-1β and chemokines possibly led to this immune cell infiltration (*Böni-Schnetzler et al., 2008*). Since it can regulate various inflammatory processes, any change in the level of IL-1β in blood or tissue possibly affects these processes. Larsen et al. suggested that blocking IL-1β is a potential therapy in the treatment of T2DM (*Larsen et al., 2007*). The *IL-1B (-511) C*

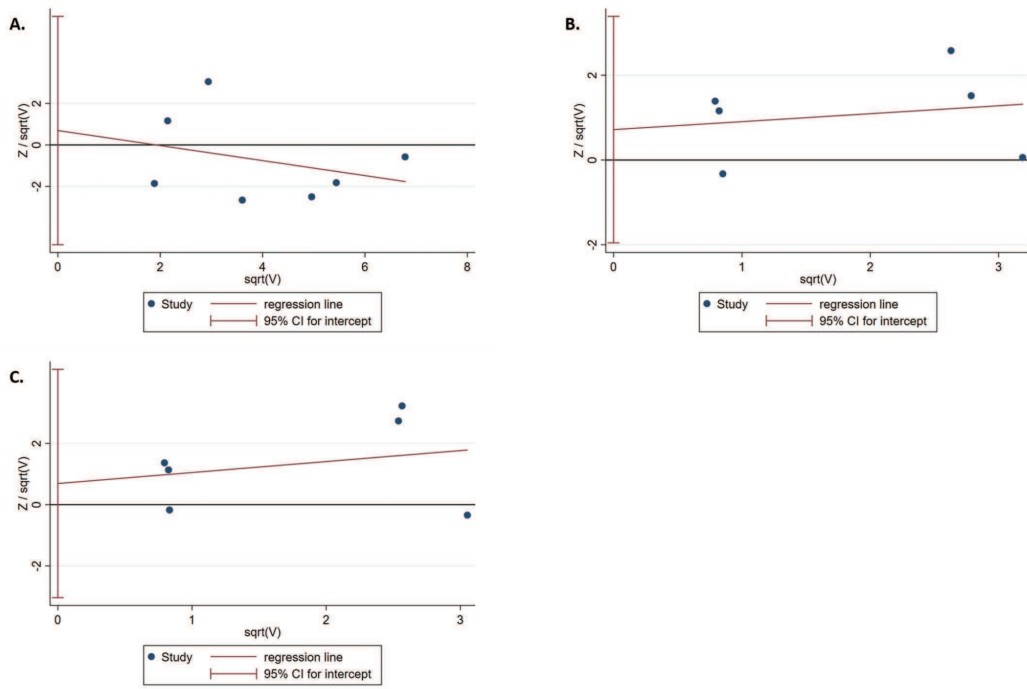

**Figure 4** **Harbord test for publication bias testing.** (A) Publication bias of *IL-1B (-511)* associated with T2DM in the dominant model (*TT + TC vs CC*); (B) Publication bias of *IL-1RN* (VNTR) associated with T2DM in the recessive model (*2\*2\* vs 2\*1\* + 1\*1\**); (C) Publication bias of *IL-1RN* (VNTR) associated with T2DM in the homozygous model(*2\*2\* vs 1\*1\**).

allele is associated with higher expression of IL-1β and with severe inflammation in the liver, while the *(-511) T* allele is associated with lower levels of IL-1β (*Hirankarn et al., 2006*; *Vishnoi et al., 2008*). In this meta-analysis, we found that the *IL-1B (-511) T* allele had a decreased T2DM risk in the EA subgroup, suggesting an influence of the *IL-1B (-511)* polymorphism system in this ethnic group.

The *IL-1RN 2\** allele (two repeats) is associated with increased IL-1Ra levels (*Danis et al., 1995*), which can compete with an inhibitor of IL-1β by binding to the IL-1 receptor. The *2\** allele significantly increases the IL-1β secretion *in vitro* and balances the expression of IL-1β and IL-1Ra (*Santtila, Savinainen & Hurme, 1998*), and their ratio (IL-1Ra/IL-1β) determines the severity of inflammation. It has been shown that the *IL-1RN 2\** allele is associated with a low ratio (IL-1Ra/IL-1β), thereby inducing a longer and more severe pro-inflammation (*Witkin, Gerber & Ledger, 2002*). The frequency of the *2\** allele is increased in inflammatory conditions or autoimmune diseases (*Tountas et al., 1999*; *Van der Paardt et al., 2002*). Consistently, this meta-analysis suggested that the *IL-1RN 2\** allele and *2\*2\** homozygotes increased T2DM risk, which indicated that the *IL-1RN 2\** allele and *2\*2\** homozygote polymorphism play critical roles in the development of T2DM.

Although we have devoted considerable efforts and resources to testing the potential association of the *IL-1B (-511)* and *IL-1RN* (VNTR) polymorphisms with T2DM risk, this analysis still has some limitations. First, significant heterogeneity was found in pooled

analyses between the *IL-1B (-511)* polymorphism and T2DM risk. Due to the limited number of included publications, we conducted further ethnic stratification analysis and meta-regression to identify the exact sources of between-study heterogeneity. The meta-regression results showed that ethnicity was not the source of heterogeneity (both values of tau$^2$ were not much different (0.34 *vs* 0.40)). Some possible relevant factors (genotyping method, sex) may lead to heterogeneity. However, subgroup analysis can significantly reduce heterogeneity. Second, the literature articles were primarily published in English or Chinese; thus, some eligible publications may be missing, causing some bias. Third, the publications included have some confounders, such as age, sex, diet, and exercise, whereas we used only the raw data to conduct pooled analysis; therefore, we were unable to control these possible confounders or test the potential gene-environment interactions. Fourth, the number of included publications was limited; thus, additional studies with large sample sizes and a wider variety of ethnicities are needed in the future to evaluate the association.

In conclusion, our meta-analysis first indicated that the *IL-1B (-511) T* allele polymorphism is associated with decreased T2DM risk in the EA population and that the *IL-1RN* 2* allele and *2*2 * homozygote polymorphism are strongly associated with increased T2DM risk. Further well-designed studies including different ethnicities with large sample sizes are needed to verify this conclusion.

## Abbreviations

| | |
|---|---|
| **T2DM** | type 2 diabetes mellitus |
| **IL-1** | Interleukin-1 |
| **IL-1Ra** | IL-1 receptor antagonist |
| **IL-1β** | interleukin-1β |
| **HWE** | Hardy–Weinberg Equilibrium |
| **OR** | odds ratios |
| **CIs** | confidence intervals |
| **EA** | East Asian |

### Funding

This work was supported by the National Natural Science Foundation of China (81870552, 81400790, 81600622, 81872096, 81571385, 91849118, 91849132, and 81600622), the National Key R&D Program of China (2018YFC2000400), Beijing Hospital Doctoral Scientific Research Foundation (BJ-2018-024), Beijing Hospital Nova Project (BJ-2018-139), and the Non-profit Central Research Institute Fund of Chinese Academy of Medical Sciences (2018RC330003). The funders had no role in study design, data collection and analysis, decision to publish, or preparation of the manuscript.

### Grant Disclosures

The following grant information was disclosed by the authors:

The National Natural Science Foundation of China: 81870552, 81400790, 81600622, 81872096, 81571385, 91849118, 91849132, 81600622.
National Key R&D Program of China: 2018YFC2000400.
Beijing Hospital Doctoral Scientific Research Foundation: BJ-2018-024.
Beijing Hospital Nova Project: BJ-2018-139.
The Non-profit Central Research Institute Fund of Chinese Academy of Medical Sciences: 2018RC330003.

## Competing Interests

The authors declare there are no competing interests.

## Author Contributions

- Juan Jiao and Zhaoping Wang conceived and designed the experiments, performed the experiments, analyzed the data, prepared figures and/or tables, authored or reviewed drafts of the paper, and approved the final draft.
- Yanfei Guo, Xiuqing Huang, Liang Sun, Xiaoquan Zhu and Qi Zhou analyzed the data, authored or reviewed drafts of the paper, and approved the final draft.
- Jie Liu and Ze Yang conceived and designed the experiments, authored or reviewed drafts of the paper, and approved the final draft.
- Xiaolin Ni and Danni Gao performed the experiments, analyzed the data, prepared figures and/or tables, authored or reviewed drafts of the paper, and approved the final draft.
- Huiping Yuan conceived and designed the experiments, prepared figures and/or tables, authored or reviewed drafts of the paper, and approved the final draft.

## Data Availability

The raw data is available in Tables 1 and 2.

## Supplemental Information

Supplemental information for this article can be found online at http://dx.doi.org/10.7717/peerj.12384#supplemental-information.

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
