# Peer review of "Association between IL-1B (-511)/IL-1RN (VNTR) polymorphisms and type 2 diabetes: a systematic review and meta-analysis"

_PeerJ, doi:10.7717/peerj.12384_

## Round 0.1 · original submission · Major Revisions

Dear Dr. Yuan,

Your manuscript entitled “Association between IL-1B (-511)/IL-1RN (VNTR) polymorphisms and type 2 diabetes: a systematic review and meta-analysis" which you submitted to PeerJ, has been reviewed by the editor and 3 experts in the field.

The recommendations were mixed. Two reviewers recommended acceptance (one "definite accept", one with "minor changes"), and one recommended rejection. I have reviewed the article myself and believe that some important criticisms raised by the Reviewer #2 need to be addressed. In particular, those on methodology, statistics and choice of references and relevant topics to be discussed. Therefore, I would be willing to reconsider if you wish to undertake major revisions and resubmit.

Please note that resubmitting your manuscript does not guarantee eventual acceptance. Since the requested changes are major, the revised manuscript will undergo a second round of review by the same reviewers. I must emphasize that the acceptability of the revision will depend upon the resolution of the points raised by each reviewer.

Sincerely yours,
Stefano Menini

·

Basic reporting

No comment

Experimental design

No comment

Validity of the findings

No comment

Additional comments

Commentaries to the Authors
This is an interesting, clear and direct study which focused on a relevant issue. The methods are appropriate, and the discussion is adequate, mainly regarding the limitations of the study, which are very important to be honestly exposed and discussed. The reading was very fluent without missing the necessary information.


Minor Changes
1 -Line 125: “In case of disagreement, it will be settled through negotiation.”. This Reviewer think this sentence sounds not much “scientific”. I suggest that authors change to terms like those used in the line 139 “…a third reviewer’s evaluation and discussed until a consistent conclusion was reached.”

2- In the legend of Figure 2, there is the necessity to explain what means: SA, SEA, EA, NA.

3- Despite there were presented in the Table 1 que QA of each study, the detailed results of the NOS of each 12 included study should be included as a Supplementary file. It is interesting to be accessed in order to see the obtained scores for each part of NOS.

4- In the Figure 1, there is the term “irrelevance”. I think it is more adequate “irrelevant”.

Reviewer 2 ·

Basic reporting

Clear, unambiguous language
• The first sentence of the abstract states that “Interleukin-1 (IL-1) plays an essential role in the immune pro-inflammatory process, inducing the development of type 2 diabetes mellitus (T2DM).” In this form, the statement appears to suggest that IL-1 levels are causal for T2D. However, T2D is a complex disease influenced by multiple environmental, ethnic and genetic factors. This sentence needs to be rephrased so that it is clear that immune inflammation is one factor among many in T2D development.
• The variant identifiers used throughout the manuscript are acceptable, however, it would be helpful for readers if a dbSNP identifier or chromosome:position coordinates were provided for the first mention of the variant, particularly in the abstract. In addition, for missense variants please indicate where in the amino acid sequence the substitution takes place, and which amino acid is substituted and what it is replaced with. E.g. Arg358Pro.
• Line 66 change “sensitive” to “sensitivity”
• Line 87: the word “epidemic” does not make sense here. Consider changing this to “prevalent”
• Line 94: add the word “pancreatic” before “β-cells”
• I disagree with the following statement (line 95): “The most typical characteristic of T2DM is chronic and low-grade inflammation”. The most typical characteristic of T2D, insulin resistance, is mentioned in the previous sentence. I do agree that inflammation is widely observed in T2D, however, it is not the most typical characteristic.
• The statement: “Many genetic association studies have been performed to estimate the relationship of the IL-1B (-511) and IL-1RN (VNTR) polymorphisms with T2DM risk” needs to reference the association studies that have investigated this relationship.


Literature references, sufficient field background/context
• Currently, the introduction outlines why inflammation is involved in T2D and what the role of IL- β in T2D may be. However, greater emphasis on findings from previous observational studies such as Spranger 2003 and genetic studies needs to be highlighted. In addition, the findings from previous genetic studies need to be discussed in more detail. i.e. Provide estimates of the association between IL-1 levels and T2D risk from several studies. Critically evaluate why previous analyses showed such contrasting results etc.
• Related to the point above, currently, there is no discussion of clinical trial data in this context. Several large clinical trials have evaluated the role of IL-1 antagonism in preventing CAD, with T2D as a secondary endpoint. E.g. CANTOS and TRACK trials. The findings from these studies need to be discussed as they are highly relevant.
• The reference provided for the following statement: “Moreover, the increased level of IL-1β in human pancreatic cells due to the elevated glucose concentrations and decreased level of IL-1 receptor antagonist (IL-1Ra) in islets of T2DM patients result in impaired insulin secretion, decreased cell proliferation, and apoptosis of β-cells.” Does not mention any of the following: impaired insulin secretion, decreased cell proliferation, and apoptosis of β-cells. Either a more suitable reference needs to be found that demonstrates these effects, or this statement needs to be amended with demonstrable findings from the literature.
• The following statement in the discussion (lines 257-260): “Eventually, it was suggested that IL-1β can induce the inflammatory microenvironment of islets, leading to impaired insulin secretion, decreased cell proliferation and apoptosis of β-cells, ultimately contributing to the development of T2DM.” references an article about IDDM which is type 1 diabetes. Therefore, this statement is factually incorrect as the reference does not concern T2D.

Professional article structure, figures, tables. Raw data shared
• When assessing the quality of the included studies using the Newcastle Ottawa Scale, the final scores for each study are included in Table 1, however, it would be helpful to include a full breakdown of the scoring for each study in a supplemental table as well.
• Table 1: T2D ascertainment in each study needs to be captured in more detail and where an explanation provided in cases where the diagnostic criteria were NA. In addition, the covariates used in each study need to be summarised.

Self-contained with relevant results to hypotheses
• The results presented are self-contained and relevant to the hypotheses, however, serious methodological issues are present (as outlined in the experimental design section) that cast doubt on the validity of the results.

Experimental design

Original primary research within scope of journal
• This study does represent original research within the scope of this journal.

Research question well defined, relevant and meaningful
• The original research question was well defined, meaningful and relevant but was not convincingly assessed due to issues with the statistical methodology.

Rigorous investigation performed to a high technical & ethical standard
• Most genetic association studies are performed using an additive model, with dominant or recessive models considered as sensitivity analyses. It is concerning that no additive model has been considered for this analysis. This needs to be added to provide estimates that are comparable with the rest of the literature. In addition, the various models used need to be substantiated in the methods.
• It is unclear whether between study heterogeneity was only considered significant when both the Cochran’s Q test and the I2 test statistics were significant. I would argue that if either were significant, one should perform a sub-group analysis.
• I disagree with using different association test methods in the presence of significant heterogeneity vs without significant heterogeneity. The authors should use a single test for association and in the case of heterogeneity, perform sub-group analyses.
• Genetic associations are known to vary widely by ancestry due to differing allele frequencies. As cohorts from East Asia, South-East Asia and North Africa have been included in this analysis, results should be stratified by ethnicity. Table 3 shows the results for “total” vs “EA” while Figure 2 shows the results within each ancestry subgroup. The association estimates for each ancestry are robust to confounding by ancestry but the total meta-analysis estimate is likely to be confounded and should therefore not be presented in Table 3. Related to this, the heterogeneity estimates for the total meta-analysis show significant heterogeneity, which is contributing to the non-significant effect of IL-1 levels in the total meta-analysis. The methods also do not mention whether stratifying by ancestry was considered. In order to accurately ascertain the association between genetically mediated IL-1 levels and T2D risk, the authors need to present their findings in Table 3 stratified by ancestry.
• It is unclear how the significance thresholds for statistically significant heterogeneity between studies were ascertained. The authors should provide an indication of whether this was determined by the number of included studies or another method.
• The sensitivity analyses described in the discussion are not mentioned in the methods section. These should be fully explained for a thorough understanding of the analyses conducted.
• The raw data from each of the publications was used in each case, without controlling for established confounders such as age, sex, lifestyle etc. While the authors acknowledge this in the discussion, it does emphasise that the estimates from this study are likely to be highly confounded and potentially invalid.

Methods described with sufficient detail & information to replicate
• The following basic reporting guidelines are followed: the authors who performed the systematic search are identified, PRISMA flow diagram is figure 1, PRISMA checklist is present. However, the authors have not provided a completed checklist outlining information about the genetic meta-analysis (genetics checklist). In addition, it is not clear where the systematic review has been registered (registration number: 268494). It is not registered at PROSPERO or CAMARADES
• It is unclear how the significance thresholds for statistically significant heterogeneity between studies were ascertained. The authors should provide an indication of whether this was determined by the number of included studies or another method.
• It is unclear from the methods section whether inverse-variance weighted meta-analysis was used to estimate the effect of IL-1 levels on T2D risk. This needs to be clarified in the methods.

Validity of the findings

As outlined in previous sections, the authors used the raw data from each of the publications to conduct their analysis. As they outline in the discussion, they did not correct for any study level covariates, leading to serious methodological questions about the validity of the results which are likely confounded. In addition, some of the statistical methodology concerning stratification by ancestry and between study heterogeneity needs to be reconsidered. All details of my suggestions are outlined above.

Additional comments

No comment

Reviewer 3 ·

Basic reporting

The article is well written and easy to read. The background and references provided are exhaustive, I found only one reference missing when commenting on the VNTRs frequency (line 107).
Figures and tables are well detailed.

Experimental design

The aim of the study is clearly stated, as well as the strategy (e.g. inclusion criteria) used to perform the analysis.
The statistical approach used seems appropriate and all the analyses performed are well described.

Validity of the findings

The results are well presented, discussion and conclusions are exhaustive. The meta-analysis performed in the present study is comprehensive, well performed from a statistical point of view, and provides new evidences in favor of the association of IL-1B and IL-1RN variants with T2DM risk.

---

## Round 0.2 · accepted · Accept

Dear Dr. Yuan,

Thank you for submitting a revised version of your manuscript. I am pleased to inform you that your manuscript is accepted for publication in PeerJ in its current form.

I thank all reviewers for their effort in improving the manuscript and the authors for their cooperation throughout the review process.
Sincerely yours,

Stefano Menini

·

Basic reporting

In this revised version, the authors had answered the questions and had corrected the mistakes. I recommend this study for publication.

Experimental design

In this revised version, the authors had answered the questions and had corrected the mistakes. I recommend this study for publication.

Validity of the findings

In this revised version, the authors had answered the questions and had corrected the mistakes. I recommend this study for publication.

Additional comments

In this revised version, the authors had answered the questions and had corrected the mistakes. I recommend this study for publication.